# Preparation, Purification, Characterization and Antioxidant Activity of Rice Bran Fermentation Broth with *Hypsizigus marmoreus*

**Yanping Chi \*, Lining Kang, Xiangying Liu, Hongrui Sun, Yue Meng, Jialin Zhang, You Kang and Yonggang Dai**

Institute of Agro-Food Technology, Jilin Academy of Agricultural Sciences (Northeast Agriculture Research Center of China), Changchun 130033, China; lnkang@sina.com (L.K.); liuxy_0621@sina.com (X.L.); hongruisun@cjaas.com (H.S.); mengyuegzl@sina.com (Y.M.); zhangjialin@cjaas.com (J.Z.); kangyou1982@hotmail.com (Y.K.); daiyg@cjaas.com (Y.D.)
\* Correspondence: chiyanpingjaas@sina.com; Tel.: +86-0431-87063424

**Abstract:** The main purpose of this study was to investigate the composition, characterization and antioxidant activity of rice bran fermentation broth, and provide a new way for high-value utilization of rice bran. Firstly, we fermented rice bran with *Hypsizigus marmoreus* and purified fermentation broth with macroporous resins. We took feruloyl oligosaccharides (FOs) concentration as the measure index, and the results showed that the maximum concentration of FOs was 0.72 mmol/L on the 6th day of rice bran fermentation. We took D101 macroporous resin as adsorption resin for rice bran fermentation broth, and the result showed that FOs concentration reached 2.38 mmol/L with the optimal purification process at pH 4.5, temperature 29 °C, ethanol concentration 55%, sample flow rate 1.5 mL/min, sample concentration 1.7 mL/min and elution flow rate 2.0 mmol/L. Secondly, the characters of rice bran fermentation broth were identified by high-performance liquid chromatography (HPLC) and Fourier-transform infrared spectroscopy (FTIR). These methods showed the presence of ferulic acid (FA), arabinose, xylose and glucose in rice bran fermentation broth. Finally, the in vitro antioxidant activities of rice bran fermentation broth were tested and the result showed that fermentation broth had good antioxidant activities and significantly improved after purification.

**Keywords:** rice bran fermentation broth; preparation; purification; characterization; antioxidant activity

## 1. Introduction

Rice bran, an important byproduct of the rice milling process, is a renewable resource with abundant yield. Rice bran contains more than 64% rice nutrients, beneficial ingredients such as proteins, polysaccharides, and oils. In addition, rice bran contains bioactive ingredients including phytosterols, polyphenols, γ-oryzanol, tocotrienols, B vitamins, minerals, and trace minerals [1]. It has gained attention from researchers due to its low cost, easy availability, nutrient-rich composition and high antioxidant potential. However, the utilization of the functional bioactive compounds is limited and fermentation is required. Several studies have proven that fermented rice bran has better functional properties and higher bioactive compounds than non-fermented rice bran. Rice bran fermented with *Lactobacillus fermentum* has significantly enhanced antioxidant activity [2]. Kataoka et al. found that rice bran fermented by *Aspergillus oryzae* could inhibit spontaneous onset of type 1 diabetes in non-obese diabetic (NOD) female mice [3]. Compared to non-fermented rice bran, the total phenolic compounds in rice bran fermented with *Rhizopus oryzae* for 120 h increased from 2.4 mg/g to 5.1 mg/g [4]. FOs can be obtained from rice bran by enzymatic hydrolysis or fermentation [5]. FOs is a sugar ester with both FA and oligosaccharide functions, which has antioxidant, blood glucose lowering, intestinal microorganism regulating, cholesterol lowering and immune improving ability, so it has wide application prospects in the fields of food and medicine [6,7].

Macroporous resin has the characteristics of large adsorption capacity, fast adsorption speed, easy desorption and stable properties. It can be reused and has a low production cost, so it is a suitable purification method for oligosaccharides. Liang et al. carried out purification and decolorization tests on five resins with different physical and chemical properties of pumpkin oligosaccharides, and the results showed that macroporous resin DM28 was a good choice for pumpkin oligosaccharide (POs) with a high decolorization rate (92.6%) and recovery rate (81.3%) [8]. Zhao et al. found that the total phenolic content of Perilla was increased from 28.00 mg/g to 66.62 mg/g by continuous use of XDA-8 macroporous resin and medium pressure liquid chromatography [9]. The purity of epigallo-catechin gallate and epicatechin gallate could reach 95.87% and 95.55% after purification by LX-20B resin and crystallization [10]. The process conditions for the production of prebiotic fructose-oligosaccharides by *Aspergillus* were optimized and purified by response surface analysis and the yield was increased by 1.97 times after optimization [11]. Le et al. studied the adsorption properties of five kinds of macroporous resins for phenolic compounds, and the results showed that XAD16 had the best performance and the highest total phenolic content [12].

At present, the rice bran research focuses more on proteins and polysaccharides analysis rather than fermentation. Microbial fermentation could degrade anti-nutritional factors such as phytic acid, trypsin inhibitors, and non-starch polysaccharides in rice bran, improve the functionality of rice bran, and expand its application areas. In this study, rice bran fermentation broth was prepared with *Hypsizigus marmoreus* and purified by the D101 macroporous resin. The composition, character and antioxidant ability of rice bran fermentation broth were analyzed, so as to lay a foundation for its further research and application. Rice bran fermentation broth could bring a series of advantages in terms of productivity, cost-effectiveness, time, and medium composition, and could be widely used in industries such as food, medicine, and cosmetics. In addition, it could also help reduce the pollution load from the environment.

## 2. Materials and Methods

### 2.1. Materials

We used rice bran (Rice Institute, Jilin Academy of Agricultural Sciences, Changchun, China), *Hypsizigus marmoreus* (Engineering Research Center of Chinese Ministry of Education for Edible and Medicinal Fungi, Jilin Agricultural University, Changchun, China), D101 resin, DM-21 resin, HPD700 resin, and DA201-B resin (Tianjin Bohong Resin Technology Co., Ltd., Tianjin, China) in this study.

All inorganic chemical reagents, ethanol, DPPH, and ABTS used in this experiment were purchased from China National Pharmaceutical Group Chemical Reagent Co., Ltd., Shanghai, China, peptone and agar were purchased from Shenggong Biotechnology (Shanghai) Co., Ltd., Shanghai, China, ferulic acid was purchased from Shanghai Aladdin Biochemical Technology Co., Ltd., Shanghai, China, acetonitrile was purchased from Merck Chemical Technology (Shanghai) Co., Ltd., Shanghai, China, and potatoes were purchased from local supermarkets.

### 2.2. Methods

2.2.1. Preparation of Rice Bran Fermentation Broth with *Hypsizigus marmoreus*

Preparation of *Hypsizigus marmoreus* Liquid

Potato Dextrose Agar (PDA) liquid medium: 200 g potatoes (boiled with distilled water for 30 min) and 15 g glucose were put into a beaker, with the volume adjusted to 1000 mL after complete dissolution.

*Hypsizigus marmoreus* activation: 1 g *Hypsizigus marmoreus* block was added to a 250 mL triangular bottle with 50 mL PDA liquid medium. It was placed into an incubator with 100 r/min rotation speed, 25 °C, fermentation for 5 d.

Rice Bran Fermented with *Hypsizigus marmoreus*

Rice bran liquid: Rice bran was mixed with methanol at a ratio of 1:10 for 1 min, centrifuged at 3000 r/min for 10 min, repeated for three times, and hot air-dried at 60 °C to obtain defatted rice bran. The defatted rice bran was prepared with distilled water at 50 g/L, adjusted pH to 5.5 with 1% (*v/v*) hydrochloric acid, 55 °C water bathing for 2 h.

Rice bran fermentation medium: glucose (15 g), lactose (15 g), peptone (1 g), $KH_2PO_4$ (1 g), $MgSO_4 \cdot 7H_2O$ (1 g) and VB (0.1 g) were added to 50 g/L rice bran liquid, adjusted pH to 5–6.

Rice bran fermentation: *Hypsizigus marmoreus* liquid (6 mL) and rice bran fermentation medium (60 mL) were put into triangular bottle (250 mL), fermented at 25 °C for 10 d with 100 r/min rotate speed.

2.2.2. Rice Bran Fermentation Broth Purification with Four Kinds of Macroporous Resin [13,14]
Pretreatment of Resin

The resin (D101, DM-21, HPD700 and DA201-B) was soaked with ethanol for 24 h, rinsed with water 4–6 times, until there was no obvious ethanol odor in the resin. It was soaked in 5% hydrochloric acid for 2 h and washed until neutral. Then, it was soaked in 2% (*w/w*) NaOH for 2 h and washed until neutral.

Calculation of Adsorption Rate, Desorption Rate and Recovery Rate

$$\text{Adsorption rate (\%)} = \frac{C_1V_1 - C_2V_2}{C_1V_1} \times 100\%$$

$$\text{Desorption rate (\%)} = \frac{C_3V_3}{C_1V_1 - C_2V_2} \times 100\%$$

$$\text{Recovery rate (\%)} = \frac{C_3V_3}{C_1V_1} \times 100\%$$

where: $C_1$—FOs concentration of the sample solution (μmol/mL), $V_1$—Volume of the sample solution (mL), $C_2$—FOs concentration of the solution after adsorption (μmol/mL), $V_2$-Volume of the solution after adsorption (mL), $C_3$—FOs concentration of the solution after desorption (μmol/mL), $V_3$—Volume of the solution after desorption (mL).

2.2.3. Rice Bran Fermentation Broth Purification with D101 Resin [15,16]
pH and Temperature

To investigate the effect of pH on purification with D101 resin, the sample solution was adsorbed by shaking (100 r/min) at pH values ranging from 2.0 to 8.0 for 24 h at 30 °C. To investigate the effect of temperature, the sample solution was adsorbed by shaking (100 r/min) at 20, 25, 30, 35, or 40 °C for 24 h in pH 4.0. After the resin adsorption was complete, the sample solution was filtrated by adding 20 mL 50% ethanol and desorbing via a shaking machine for 4 h at room temperature. Thereafter, the FOs content in the residual adsorption solution and desorption solution was determined, and the adsorption rate and desorption rate of the resin were calculated.

Ethanol Concentration

To investigate the effect of ethanol, the sample solution was adsorbed by shaking (100 r/min) at 25 °C for 24 h in pH 4.0. After complete adsorption, the resin was filtered and 20 mL ethanol with different concentration (*v/v*) (10%, 30%, 50%, 70%, 90%) was added. The operation was the same as 2.2.3.1., and the desorption rates and recovery rate of the resin were calculated.

Sample Flow Rate and Sample Concentration

To investigate the effect of sample flow rate, the sample solution of 1.5 mmol/L was adsorbed at the flow rates of 0.5, 1.0, 1.5, 2.0 and 2.5 mL/min, then washed with distilled water and eluted with 50% ethanol. To investigate the effect of sample concentration, samples with different concentrations (0.5, 1.0, 1.5, 2.0, 2.5 mmol/L) were loaded and adsorbed at 1.5 mL/min sample flow rate, and then washed with distilled water and eluted with 50% ethanol to determine the recovery rate.

Elution Flow Rate

The resin was loaded wet, and 1.5 mmol/L sample solution was loaded and adsorbed at a flow rate of 1.5 mL/min. After saturation, the sample solution was washed with distilled water, and then eluted with 50% ethanol solution at a flow rate of 0.5, 1.0, 2.0 and 3.0 mL/min, respectively. FOs concentration in different volumes of eluted effluents was determined (measured every 10 mL of effluents).

Orthogonal Optimization Design

According to the results of the single factor test, taking pH, temperature, ethanol concentration, sample flow rate, sample concentration and elution flow rate as the investigation factors, the orthogonal experiment of $L_{25}$ ($5^6$) six factors and five levels was carried out on rice bran fermentation broth, and FOs content was taken as the investigation index (Table 1).

**Table 1.** $L_{25}$ ($5^6$) orthogonal experimental factor level.

| No. | | | | Factors | | |
|---|---|---|---|---|---|---|
| | **A** pH | **B** Temperature/°C | **C** Ethanol Concentration /% | **D** Sample Flow Rate /(mL/min) | **E** Sample Concentration /(mmol/L) | **F** Elution Flow Rate /(mL/min) |
| 1 | 3.0 | 21 | 40 | 1.1 | 1.1 | 1.6 |
| 2 | 3.5 | 23 | 45 | 1.3 | 1.3 | 1.8 |
| 3 | 4.0 | 25 | 50 | 1.5 | 1.5 | 2.0 |
| 4 | 4.5 | 27 | 55 | 1.7 | 1.7 | 2.2 |
| 5 | 5.0 | 29 | 60 | 1.9 | 1.9 | 2.4 |

### 2.2.4. Determination of FOs [17,18]

First, 0.1 mL of fermentation dilution was mixed with 0.9 mL of 0.1 mol/L bora-glycine buffer solution (pH10), and the absorbance was measured at wavelengths of 345 nm and 375 nm. According to the molar absorbance coefficient of FA ($M^{-1}cm^{-1}$): $\varepsilon^1_{345}$ =19,662, $\varepsilon^1_{375}$ = 7630 and FOs molar absorbance coefficient ($M^{-1}cm^{-1}$): $\varepsilon^2_{345}$ = 23,064, $\varepsilon^2_{375}$ = 31,430 to calculate the concentration of FOs.

$$A_{345} = 19{,}662bC_1 + 23{,}064bC_2 \quad A_{375} = 7630bC_1 + 31{,}430bC_2$$

where: $C_1$—FA concentration (mol/L), the concentration of $C_2$—FOs (mol/L), b—cuvette thickness (cm), $A_{345}$—OD of wavelength 345 nm, $A_{375}$—OD of 375 nm wavelength

$$FOs\ (mmol/L) = [(A_{375}\varepsilon^1_{345}) - (\varepsilon^1_{375} A_{345})]/[(\varepsilon^2_{375}\varepsilon^1_{345}) - (\varepsilon^2_{345}\varepsilon^1_{375})]$$

### 2.2.5. Determination of FA [19,20]

Sample treatment: 1 mL of purified sample solution was added with 1 mL of sodium hydroxide solution with a concentration of 1 mol/L, which was hydrolyzed at 100 °C for 90 min to enable rice bran fermentation broth hydrolysis, and then neutralized with 1 M hydrochloric acid after cooling.

The HPLC conditions for the determination of FA were as follows: AgilentTC-C18 column, Agilent, Palo Alto, CA, USA, 25 °C, HPLC separation, UV detector, Agilent, Palo

Alto, CA, USA. The eluents were 5% trifluoroacetic acid (*v/v*) (A) and acetonitrile (B) at a flow rate of 0.6 mL/min in a gradient elution sequence (0 min: Solvent A 95%, Solvent B 5%; 15 min: Solvent A 80%, Solvent B 20%; 40 min: Solvent A 60%, Solvent B 40%). The injection volume was 10 μL, and the detection wavelength was 325 nm.

2.2.6. Determination of Monosaccharide Composition in Rice Bran Fermentation Broth [21,22]

Preparation of Polysaccharide Sample Solution

First, after freeze-drying, a 100 mg sample was weighed and placed in a stopper tube, and 2 mL sulfuric acid solution of 2 mol/L was added. After sealing the tube, it was hydrolyzed in a water bath at 100 °C for 2 h, cooled to room temperature, neutralized with 4 mol/L NaOH to pH 7.0, diluted with distilled water to 16 mL, and centrifuged at 1000 r/min for 5 min. Sample hydrolysate was obtained from the supernatant and used for PMP derivatization.

Derivatization of Samples

First, 250 μL of monosaccharide was taken, mixed with solution and sample solution, 250 μL of 0.5 mol/L PMP methanol solution was added, then 250 μL of 0.3 mol/L NaOH solution was added. It was mixed for 30 s on the vortex device, processed via derivatization reaction in 70 °C water bath for 1.5 h and cooled. The neutralized solution was neutralized by adding 250 μL of 0.3 mol/L HCl solution, and the neutralized solution was extracted by adding 1.5 mL of water and 5 mL of chloroform. The chloroform layer was discarded. The water layer was extracted three times, and the water layer was filtered with 0.45 μm microporous membrane before use.

Chromatographic Conditions

Chromatographic column: Agilent Eclipse X D B-C18 column (150 mm × 4.6 mm, 5 μm); mobile phase A consisted of 0.02 mol/L phosphate buffer solution (pH 6.0), and phase B consisted of acetonitrile. Elution gradient (0 min, 0% B; 10 min, 18% B; 20 min, 25% B); column temperature 30 °C. The injection volume was 10 μL, flow rate 1.0 mL/min, detection wavelength: 250 nm.

2.2.7. Determination of Rice Bran Fermentation Broth by FTIR [23]

The FTIR analysis was conducted in triplicate for each sample by an FTIR spectrometer (VERTEX 70, Bruker Inc., Karlsruhe, Germany). Initially, 1 mg$^{-2}$ mg of purified freeze-dried rice bran fermentation broth sample and a proper amount of dried KBr powder were weighed and ground in an agate mortar under an infrared lamp until it was thoroughly ground and mixed. The powder was not evenly put into the mold press, and after vacuuming, the transparent thin section was obtained by pressure. The thin section was placed on the sample shelf for infrared spectrum test, and the scanning range was 400–4000 cm$^{-1}$.

2.2.8. Antioxidant Activity [24–28]

The antioxidant capacity of rice bran fermentation broth before and after purification was determined.

Reducing Power

First, 0.2 mol/L phosphate buffer solution (pH 6.6): 1.95 g $NaH_2PO_4 \cdot 2H_2O$ and 2.68 g $Na_2HPO_4 \cdot 12H_2O$ were dissolved in 100 mL distilled water. The 1.0% potassium ferrocyanide solution with 0.5 g potassium ferrocyanide was dissolved in 50 mL distilled water. The 10% trichloroacetic acid solution with 5 g trichloroacetic acid was dissolved in 50 mL distilled water. The 0.1% ferric chloride solution with 0.05 g ferric chloride was dissolved in 50 mL distilled water. Next, 2.5 mL phosphate buffer, 2.5 mL potassium ferrocyanide and 1 mL rice bran fermentation broth were added to the test tube, well mixed. The solution was placed in a 50 °C water bath for 20 min and 2.5 mL trichloroacetic acid

was added. The solution was mixed well, centrifuged at 3000 r/min for 10 min. Next, 100 μL supernatant, 100 μL distilled water and 20 μL $FeCl_3$ solution were added to a 96-well plate, mixed well and allowed to stand for 10 min. The absorbance value $A_1$ was measured at 700 nm. The absorbance value $A_2$ of distilled water was measured for the rice bran fermentation broth solution. The formula for calculating the total reducing force (W) is as follows:

$$W = A_1 - A_2$$

Scavenging Capacity of ·OH

First, 2 mL of sample solutions with concentrations of 0.1 mg/mL, 0.2 mg/mL, 0.4 mg/mL, 0.6 mg/mL, 0.8 mg/mL and 1 mg/mL, 0.5 mL of 9 mmol/L salicylic acid-ethanol solution and 0.5 mL of 9 mmol/L $FeCl_2$ solution were added to the 10 mL stopper tube, 6.5 mL deionized water and 0.5 mL 8.8 mmol/L $H_2O_2$ solution were mixed evenly and reacted in 37 °C water bath for 1 h. We measured the absorbance value A1 at 510 nm; the absorbance values of distilled water replacing $H_2O_2$ solution and rice bran fermentation broth solution were $A_2$ and $A_0$. The formula for calculating the ·OH clearance rate (Z, %) is as follows.

$$Z = (1 - \frac{A_1 - A_2}{A_0}) \times 100$$

Scavenging Capacity of $ABTS^+$·

We prepared 7 mmol/L ABTS solution and 2.45 mmol/L potassium persulfate solution. The ABTS solution and potassium persulfate solution were mixed in a 1:1 (*v/v*) ratio to form the ABTS stock solution. We diluted the ABTS stock solution with distilled water to obtain ABTS working solution with an absorbance value of $0.7 \pm 0.02$ at 734 nm. We added 10 μL rice bran fermentation broth solution and 200 μL ABTS working solution to a 96-well plate and allowed it to stand for 6 min at room temperature. We measured the absorbance value $A_1$ at 734 nm. The absorbance of distilled water instead of ABTS working solution and rice bran fermentation broth solution were $A_2$ and $A_0$. The formula for calculating ABTS+· clearance rate (Y, %) is as follows.

$$Y = (1 - \frac{A_1 - A_2}{A_0}) \times 100$$

Scavenging Ability of DPPH·

First, 3.9432 mg DPPH was dissolved in 50 mL anhydrous ethanol solution to prepare a DPPH solution (0.2 mmol/L), and it was stored at 4 °C in the dark. We added 40 μL rice bran fermentation broth solution and 200 μL DPPH solution to a 96-well plate, mixed it well and allowed it to stand for 30 min in a dark environment. We measured the absorbance value $A_1$ at 517 nm. The absorbance of anhydrous ethanol replacing DPPH solution was $A_2$. The absorbance value of distilled water instead of rice bran fermentation broth solution was $A_0$. The formula for calculating DPPH· clearance rate (X, %) is as follows.

$$X = (1 - \frac{A_1 - A_2}{A_0}) \times 100$$

where: $A_1$—absorbance of sample solution; $A_2$—absorbance of 95% ethanol instead of DPPH solution; $A_0$—absorbance of deionized water instead of sample solution.

2.2.9. Statistical Analysis

Each index of the sample was set three times in duplicate. IBM SPSS Statistics 25.0 software was used for Duncan's multiple comparison analysis, Origin 95C was used for plotting, and the results were expressed as mean (Means) $\pm$ standard deviation (SD). Statistical results with $p < 0.05$ indicated that the difference reached the significant level.

## 3. Results

### 3.1. Preparation of Rice Bran Fermentation Broth with Hypsizigus marmoreus

Rice bran was fermented for 10 d with *Hypsizigus marmoreus*, the changes in FOs content in the fermentation broth are shown in Figure 1. With the prolongation of fermentation time, FOs content in fermentation broth gradually increased, and reached the highest level of 0.72 mmol/L on the sixth day, and then decreased. In the early stage of fermentation, various enzymes produced by the *Hypsizigus marmoreus* gradually accumulated with sufficient nutrition, and FOs concentration increased with the enzyme activities enhanced. In the later stage of fermentation, there was a lack of nutrients, and changes in the growth environment. The bacterial body enters the decay phase, enzyme activities weakened, metabolic products no longer accumulated, and FOs content decreased. So, after 6 days of fermentation, the FOs content gradually decreased.

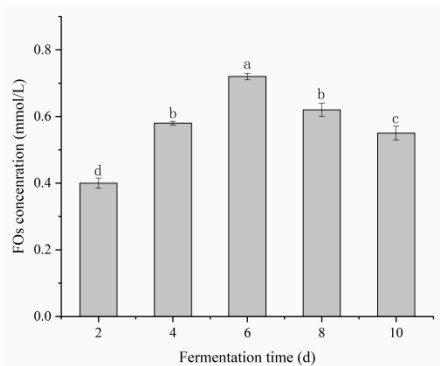

**Figure 1.** FOs content during rice bran fermentation with *Hypsizigus marmoreus*. Different letters indicate significant differences ($p < 0.05$).

### 3.2. Purification of Rice Bran Rice Bran Fermentation Broth

3.2.1. Screening of Macroporous Resin

Purification Parameters of Four Macroporous Resins

The adsorption and desorption rates of non-polar resins (D101 and HPD700) were higher than those of medium-polar resins (DM-21) and polar resins (DA201-B) (Table 2). The D101 macroporous resin has stronger adsorption and desorption capacity for rice bran fermentation broth than the other three resins. Therefore, D101 macroporous resin was selected to purify rice bran fermentation broth.

**Table 2.** Physical parameters, adsorption rate and desorption rate of four macroporous resins.

| Resin Type | Polarity | The Average Pore Diameter/μm | Adsorption Rate/% | Desorption Rate/% |
|---|---|---|---|---|
| D101 | non-polar | 90~100 | 96.60 ± 0.30 [a] | 98.81 ± 0.20 [a] |
| HPD700 | non-polar | 90~100 | 86.54 ± 0.24 [b] | 90.47 ± 0.25 [b] |
| DA201-B | polar | 90~100 | 72.36 ± 0.53 [c] | 82.43 ± 0.41 [c] |
| DM-21 | neutral | 130~140 | 85.10 ± 0.26 [d] | 87.97 ± 0.15 [d] |

Different letters in the same column indicate significant differences ($p < 0.05$).

3.2.2. Single Factor Experiments on Purification of D101 Macroporous Resin

pH and Temperature

The adsorption and desorption rates increased and then decreased with the increase in pH value, and reached the maximum at pH 4.0, which were 90.65% and 95.52%, respectively (Figure 2a). With the increase in temperature, the adsorption and desorption rates gradually decreased and reached the maximum at 25 °C, which were 89.64% and 95.23%, respectively (Figure 2b).

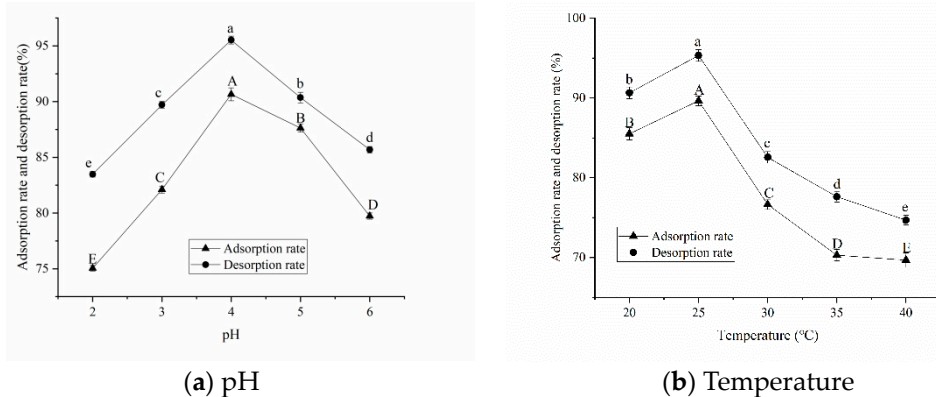

**(a)** pH                                        **(b)** Temperature

**Figure 2.** Effects of pH (**a**) and temperature (**b**) of rice bran fermentation broth purification on purification by D101 macroporous resin. Different letters in the same curve indicate significant differences ($p < 0.05$).

Ethanol Concentration

The effect of ethanol concentration desorption rate showed that with the increase in ethanol concentration, the desorption rate firstly increased and then decreased. The desorption rate reached 93.70% at an ethanol volume fraction of 50% (Figure 3).

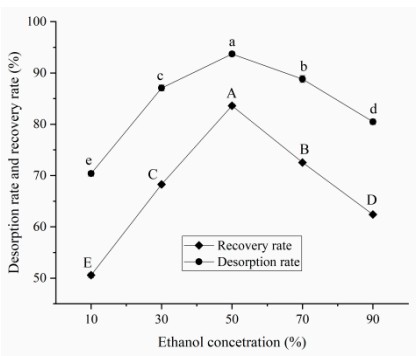

**Figure 3.** Effects of ethanol concentration of rice bran fermentation broth purification on purification by D101 macroporous resin. Different letters in the same curve indicate significant differences ($p < 0.05$).

Sample Flow Rate and Sample Concentration

With the increase in sample flow rate, the recovery firstly increased and then decreased. When the sample flow rate was 1.5 mL/min, the recovery reached 90.98% (Figure 4a). When the sample concentration was 1.5 mmol/L, the maximum recovery was 89.32% (Figure 4b).

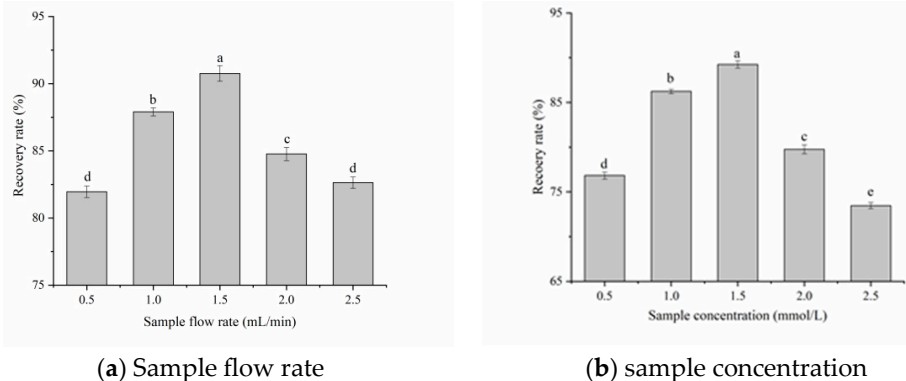

**(a)** Sample flow rate                        **(b)** sample concentration

**Figure 4.** Effects of sample flow rate (**a**) and sample concentration (**b**) of rice bran fermentation broth purification on purification by D101 macroporous resin. Different letters indicate significant differences ($p < 0.05$).

Elution Flow Rate and Eluent Volume

With the increase in eluent volume, FOs content in the eluent first increased and then decreased (Figure 5). When the eluent volume was 10–30 mL, FOs content in the eluent increased sharply and reached the peak at 30 mL. The elution flow rate was 1.5 mL/min, FOs contents was higher than the other. FOs content was 1.83 mmol/L with 30 mL eluent volume and 1.5 mL/min elution flow rate.

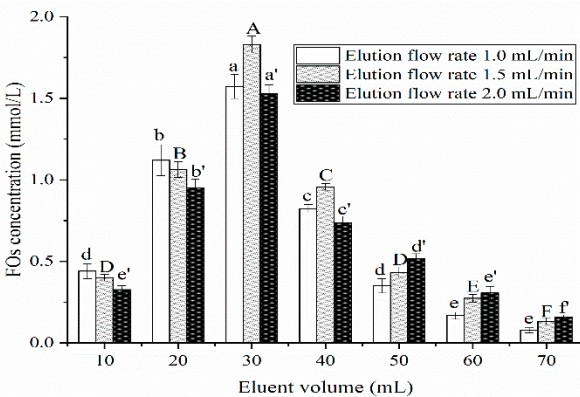

**Figure 5.** Effects of elution flow rate and eluent volume of rice bran fermentation broth purification on purification by D101 macroporous resin. Different letters in the same bar chart indicate significant differences ($p < 0.05$).

### 3.2.3. Orthogonal Test Results

The orthogonal test results are shown in Table 3, where k is the mean ($K_1$ is the average value of FOs content with five level 1 at the corresponding factor, $K_2$ is the average value of FOs content with five level 2 at the corresponding factor, and so on.), R is the range. According to the R value, the order of importance of the six factors affecting rice bran fermentation broth purification are elution flow rate, pH, sample concentration, ethanol concentration, sample flow rate and temperature. From Table 3, according to the analysis of mean, we chose the level with the highest mean for each factor as the optimal level. Through our analysis, we can conclude that the optimal scheme was $A_4B_5C_4D_3E_4F_3$. According to the experimental results, we could see that the FOs content of $A_4B_1C_4D_2E_5F_3$ and $A_4B_5C_3D_1E_4F_2$ were very close in Table 3, and higher than another schemes. Thus, we verified the scheme ($A_4B_5C_4D_3E_4F_3$) of orthogonal optimization and the second scheme ($A_4B_1C_4D_2E_5F_3$, $A_4B_5C_3D_1E_4F_2$) with higher FOs contents through experiments; the results are shown in Table 4. According to Table 4, the FOs concentration of $A_4B_5C_4D_3E_4F_3$ scheme was the highest, and we concluded that the $A_4B_5C_4D_3E_4F_3$ scheme was the optimal scheme (pH 4.5, temperature 29 °C, ethanol concentration 55%, sample flow rate 1.5 mL/min, sample concentration 1.7 mmol/L, elution flow rate 2.0 mL/min). Under these conditions, the FOs concentration of rice bran fermentation broth could reach 2.38 mmol/L.

**Table 3.** Results and analysis of orthogonal test.

| No. | Factors | | | | | | FOs Content /(mmol/L) |
|---|---|---|---|---|---|---|---|
| | **A** pH | **B** Temperature | **C** Ethanol Concentration | **D** Sample Flow Rate | **E** Sample Concentration | **F** Elution | |
| 1 | 1 | 1 | 1 | 1 | 1 | 1 | $1.02 \pm 0.025$ [q] |
| 2 | 1 | 2 | 2 | 2 | 2 | 2 | $1.28 \pm 0.015$ [no] |
| 3 | 1 | 3 | 3 | 3 | 3 | 3 | $1.96 \pm 0.0057$ [b] |
| 4 | 1 | 4 | 4 | 4 | 4 | 4 | $1.80 \pm 1.015$ [fg] |
| 5 | 1 | 5 | 5 | 5 | 5 | 5 | $1.55 \pm 0.010$ [j] |
| 6 | 2 | 1 | 2 | 3 | 4 | 5 | $1.58 \pm 0.0057$ [j] |
| 7 | 2 | 2 | 3 | 4 | 5 | 1 | $1.30 \pm 0.015$ [n] |

**Table 3.** *Cont.*

| No. | Factors | | | | | | FOs Content /(mmol/L) |
|---|---|---|---|---|---|---|---|
| | A pH | B Temperature | C Ethanol Concentration | D Sample Flow Rate | E Sample Concentration | F Elution | |
| 8 | 2 | 3 | 4 | 5 | 1 | 2 | $1.44 \pm 0.0057$ [kl] |
| 9 | 2 | 4 | 5 | 1 | 2 | 3 | $1.89 \pm 0.035$ [d] |
| 10 | 2 | 5 | 1 | 2 | 3 | 4 | $1.68 \pm 0.015$ [i] |
| 11 | 3 | 1 | 3 | 5 | 2 | 4 | $1.78 \pm 0057$ [g] |
| 12 | 3 | 2 | 4 | 1 | 3 | 5 | $1.72 \pm 0.015$ [h] |
| 13 | 3 | 3 | 5 | 2 | 4 | 1 | $1.68 \pm 0.010$ [i] |
| 14 | 3 | 4 | 1 | 3 | 5 | 2 | $1.82 \pm 0.015$ [ef] |
| 15 | 3 | 5 | 2 | 4 | 1 | 3 | $1.85 \pm 0.015$ [e] |
| 16 | 4 | 1 | 4 | 2 | 5 | 3 | $2.20 \pm 0.015$ [a] |
| 17 | 4 | 2 | 5 | 3 | 1 | 4 | $1.92 \pm 0.057$ [c] |
| 18 | 4 | 3 | 1 | 4 | 2 | 5 | $1.25 \pm 0.057$ [o] |
| 19 | 4 | 4 | 2 | 5 | 3 | 1 | $1.28 \pm 0.010$ [n] |
| 20 | 4 | 5 | 3 | 1 | 4 | 2 | $2.21 \pm 0.015$ [a] |
| 21 | 5 | 1 | 5 | 4 | 3 | 2 | $1.45 \pm 0.0057$ [k] |
| 22 | 5 | 2 | 1 | 5 | 4 | 3 | $1.72 \pm 0.015$ [h] |
| 23 | 5 | 3 | 2 | 1 | 5 | 4 | $1.42 \pm 0.010$ [l] |
| 24 | 5 | 4 | 3 | 2 | 1 | 5 | $1.10 \pm 0.035$ [p] |
| 25 | 5 | 5 | 4 | 3 | 2 | 1 | $1.36 \pm 0.010$ [m] |
| $k_1$ | 1.522 | 1.606 | 1.498 | 1.652 | 1.466 | 1.328 | |
| $k_2$ | 1.578 | 1.588 | 1.482 | 1.588 | 1.512 | 1.498 | |
| $k_3$ | 1.770 | 1.550 | 1.670 | 1.728 | 1.618 | 1.972 | |
| $k_4$ | 1.772 | 1.578 | 1.704 | 1.530 | 1.798 | 1.720 | |
| $k_5$ | 1.140 | 1.730 | 1.698 | 1.554 | 1.658 | 1.440 | |
| R | 0.362 | 0.180 | 0.222 | 0.198 | 0.332 | 0.644 | |
| optimum | $A_4$ | $B_5$ | $C_4$ | $D_3$ | $E_4$ | $F_3$ | |

Different letters in the same column indicate significant differences ($p < 0.05$).

**Table 4.** Verification experiment results.

| Number | Experimental Scheme | FOs Content |
|---|---|---|
| 1 | $A_4B_5C_4D_3E_4F_3$ | $2.38 \pm 0.0153$ [a] |
| 2 | $A_4B_5C_3D_1E_4F_2$ | $2.26 \pm 0.0200$ [b] |
| 3 | $A_4B_1C_4D_2E_5F_3$ | $2.24 \pm 0.0100$ [b] |

Different letters in the same column indicate significant differences ($p < 0.05$).

### 3.3. Analysis of Rice Bran Fermentation Broth Composition

The characteristic peak time of FA standard was 12.475 min in Figure 6a, and that of rice bran fermentation broth was 12.516 min in Figure 6b. The peak times of Figure 6a,b was relatively consistent, indicating the presence of FA in rice bran fermentation broth.

Compared with Figure 6c,d, it was found that the peak time of rice bran fermentation broth and monosaccharide standard mixture were similar. The peak time of the derivative reagent was about 14.8 min, and the peak time was 19.8 min, 20.3 min and 20.9 min, which were glucose, arabinose and xylose, respectively.

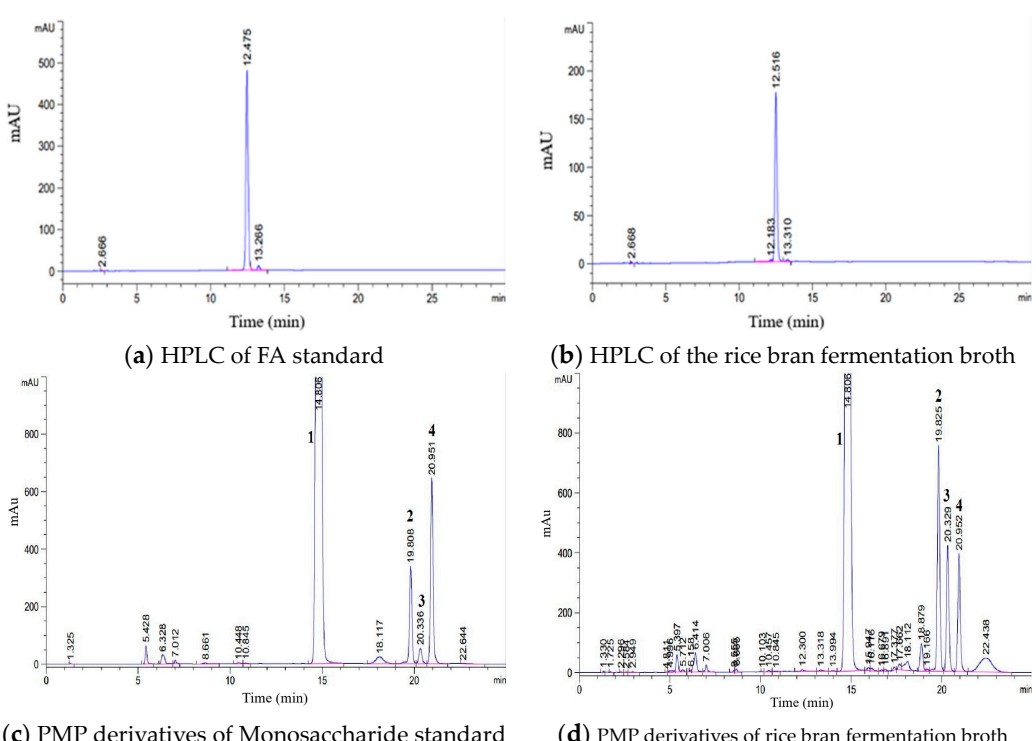

(**a**) HPLC of FA standard

(**b**) HPLC of the rice bran fermentation broth

(**c**) PMP derivatives of Monosaccharide standard

(**d**) PMP derivatives of rice bran fermentation broth

**Figure 6.** HPLC of rice bran fermentation broth FA and monosaccharide. 1. PMP derivatives; 2. glucose; 3. arabinose; 4. xylose.

### 3.4. Infrared Spectra of Rice Bran Fermentation Broth

The infrared spectrum of rice bran fermentation broth is shown in Figure 7. The figure shows that the strong and broad absorption peak of 3415.78 $cm^{-1}$ is in the range of 3000–3750 $cm^{-1}$, which belongs to the stretching vibration region of C-H association, indicating the existence of oligosaccharide association between light groups and monosaccharide hydroxyl hydrogen bonds. The absorption peak of 2931.67 $cm^{-1}$ is in the range of 2700–3000 $cm^{-1}$, which belongs to the saturated C-H stretching vibration region, indicating the existence of methyl and carboxyl groups. The absorption peak of 1650.99 $cm^{-1}$ is in the range of 1650–1900 $cm^{-1}$, which belongs to the stretching vibration zone of C=O and C=C, indicating the existence of esters and alkenes. The absorption peak of 1411.83 $cm^{-1}$ is in the range of 1300–1475 $cm^{-1}$, which belongs to the flexural vibration region in the C-H plane, indicating the presence of carboxylate in the fermentation broth. The absorption peaks of 1251.75 $cm^{-1}$, 1151.45 $cm^{-1}$, 1080.08 $cm^{-1}$ and 1028.01 $cm^{-1}$ are in the range of 1000–1300 $cm^{-1}$, which belong to the stretching vibration region of C-O, indicating the existence of ester bonds or esters. The absorption peak of 891.07 $cm^{-1}$ is in the range of 650–1000 $cm^{-1}$, which belongs to the unsaturated C-H out-of-plane bending vibration region, indicating the existence of B-Glycosidic bond. According to the above analysis, there was esterified FA in rice bran fermentation broth, which is a sugar substance containing an ester bond.

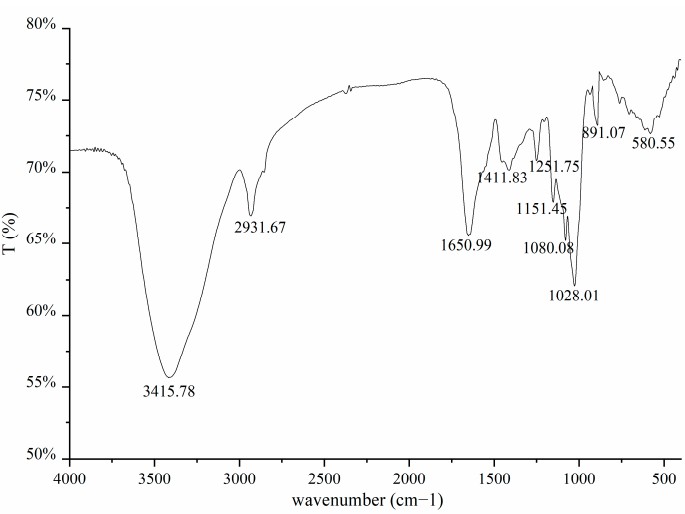

**Figure 7.** FTIR of rice bran fermentation broth.

### 3.5. Antioxidant Activity of Rice Bran Fermentation Broth

The reducing power of rice bran fermentation broth after purification was enhanced, and was 2.18 times that of rice bran fermentation broth without purification (Figure 8a). The scavenging rate of Hydroxyl radical increased from 43.12% to 85.43%, which was 1.98 times higher than that when unpurified (Figure 8b). The scavenging ability of purified rice bran fermentation broth on ABTS radical was significantly enhanced; the scavenging rate increased from 31.42% to 70.29%, and the scavenging ability was 2.24 times that when unpurified (Figure 8c). Purified rice bran fermentation broth had an excellent scavenging ability on DPPH radical, where the scavenging rate increased from 31.12% to 92.46%, and the scavenging ability was 2.94 times that when unpurified (Figure 8d).

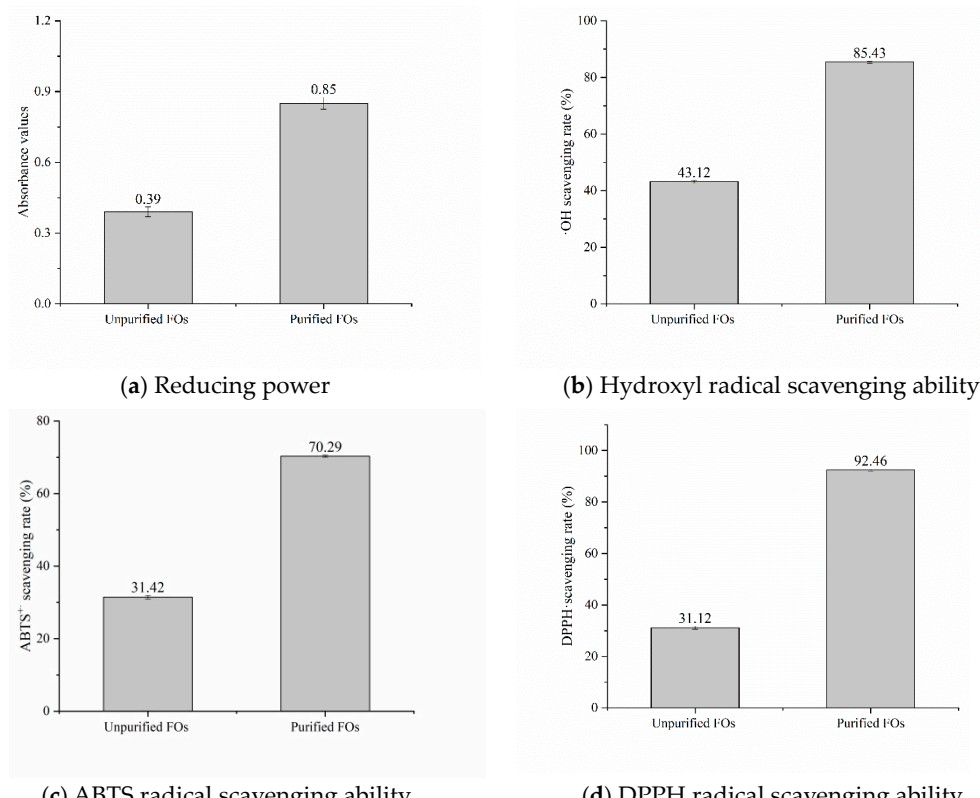

(**a**) Reducing power

(**b**) Hydroxyl radical scavenging ability

(**c**) ABTS radical scavenging ability

(**d**) DPPH radical scavenging ability

**Figure 8.** Antioxidant activity of rice bran fermentation broth.

## 4. Discussion

The fermentation of edible fungi could produce amylase, protease, cellulase, and xylanase and FOs were generated from rice bran under the joint action of several enzymes [29,30]. In this study, we fermented rice bran with *Hypsizigus marmoreus*. The FOs concentration of rice bran fermentation broth was 0.72 mmol/L. The FOs content in rice bran fermentation broth was relatively low. This would limit the application of rice bran. We must find a way to increase the concentration of FOs. Using resin for separating and purifying was an effective method. Macroporous resins have the advantages of high selectivity, fast adsorption speed, low cost, less solvent consumption and easy regeneration, and were widely used in purification of functional oligosaccharides [31,32]. In this study, the D101 macroporous resins were suitable for purification of rice bran fermentation broth, by single factor and orthogonal experiment experiments, FOs concentration was increased by 2.30 times from 0.72 mmol/L to 2.38 mmol/L. Although the concentration of purified FOs increased by 2.30 times, the purification effect was not significant. We could use two types of resin combination or ultrasound-assisted resin purification methods to improve purification efficiency. Zhang et al. used ultrasound-assisted LK825 macroporous resin to purify anthocyanins from blueberry pomace, and the purified anthocyanin contents increased by 12.8 times [33]. Mi et al. successively used macroporous resin medium pressure preparative chromatography and gel resin medium pressure preparative chromatography for separation and purification, and the contents of two anthocyanin components of Lycium barbarum were 3.62 and 1.65 times higher than that of macroporous resin medium pressure preparative chromatography alone [34]. We could also ferment rice bran with another edible or probiotic to increase the contents of FOs in the fermentation broth.

Rice bran fermentation broth contained FA, arabinose, xylose and glucose by HPLC. The FTIR showed that there were associated hydroxyl group, carboxyl group, C=C, trisubstituted benzene ring, ester bond and β-type glycosidic bonds in rice bran fermentation. The existence of these made it have better functional properties. FA is a phenylpropionate compound, which has natural antioxidant activity, and its physiological activity can be improved by modifying polysaccharides with FA [35]. FOs are compounds formed by the combination of FA and oligosaccharides through ester bonds, so they have good antioxidant activity.

After fermentation and purification, the antioxidant ability of rice bran fermentation broth was significantly increased, this was consistent with the study of Ardiansyah et al. in comparing non-volatile compounds of fermented and non-fermented Inpari 30 and Cempo Ireng rice bran and its blood pressure-lowering activity [36]. The studies of Alejandra B. Omarini et al. fermented rice bran with *Pleurotus sapidus*, and the result showed that an improvement was found in the nutritional quality of rice bran after fermentation with *Pleurotus sapidus*, since protein, carbohydrates, minerals, and specific fatty acid components were increased [37].

## 5. Conclusions

Rice bran fermentation broth with *Hypsizigus marmoreus* contains feruloyl oligosaccharides and FA. The concentration of feruloyl oligosaccharides purified By D101 macroporous resin increased from 0.72 mmol/L to 2.38 mmol/L; an increase of 2.3 times. The reducing power, Hydroxyl radical scavenging ability, ABTS radical scavenging ability and DPPH radical scavenging ability of rice bran fermentation broth were significantly increased after purification.

**Author Contributions:** Conceptualization: Y.C.; methodology: Y.C., Y.D., L.K. and H.S. and Y.M.; data curation: Y.C., Y.K., J.Z. and X.L.; writing—original draft preparation: Y.C.; writing—review and editing: Y.C.; project administration: Y.C. and Y.D.; funding acquisition: Y.C. All authors have read and agreed to the published version of the manuscript.

**Funding:** This research was funded by Key Research and Development Projects of Jilin Province, grant number 20230203182SF.

**Institutional Review Board Statement:** Not applicable.

**Informed Consent Statement:** Not applicable.

**Data Availability Statement:** The raw data supporting the conclusions of this article will be made available by the authors on request.

**Conflicts of Interest:** The authors declare no conflicts of interest.

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
