# Peer review of "Preparation, Purification, Characterization and Antioxidant Activity of Rice Bran Fermentation Broth with Hypsizigus marmoreus"

_fermentation, doi:10.3390/fermentation10040188_

Round 1
Reviewer 1 Report
Comments and Suggestions for Authors
The manuscript entitled "Preparation, purification, characterization and antioxidant activity of rice bran fermentation broth with Hypsizigus marmoreus" by Chi et al., describe a method to generate fermentation broth from rice bran. The authors also described about the composition and functional activity of the rice bran fermentation broth.
Specific Concerns:
1. Scientific nomenclature should adhere to guidelines.
2. Abbreviation should be provided in each case.
3. Axis labels are missing in case of Figure 3.
4. Does the output of the current method able to detect, discriminate and scavenge different oxidative radicals such as hydroxyl radical, superoxide radicals, peroxyl radicals and other reactive oxygen species and reactive nitrogen species?
5. A brief note on the limitations of the study might be helpful.
6. What are the significant advantages of the current method compared to the other reported methods.
7. Describe about the applications of the study.
Comments on the Quality of English Language
There are several grammatical errors, typographical errors and incomplete sentences throughout the manuscript that requires attention.
Reviewer 2 Report
Comments and Suggestions for Authors
This paper investigates the composition, characterization, and antioxidant activity of rice bran fermentation broth. The study involves several key steps:
Ø Fermentation of rice bran using Hypsizigus marmoreu.
Ø Purification of the fermentation broth using macroporous resins, with a focus on feruloyl oligosaccharides (FOs) concentration as the measure index.
Ø Identification of the characteristics of the rice bran fermentation broth using high-performance liquid chromatography (HPLC) and infrared spectroscopy, revealing the presence of ferulic acid, arabinose, xylose, and glucose, which are potentially beneficial for health.
Ø Evaluation of the antioxidant activity of the rice bran fermentation broth through in vitro tests, demonstrating its good antioxidant activity, which further improves after purification.
Overall, the study proposes a new method for the high-value utilization of rice bran by fermenting it and purifying the fermentation broth, highlighting its potential health benefits due to the presence of specific components and its antioxidant activity.
According to the results of the single-factor test, an orthogonal experiment of L25 (56) with 6 factors and 5 levels was conducted on rice bran fermentation broth, considering pH, temperature, ethanol concentration, sample flow rate, sample concentration, and elution flow rate as the investigation factors, with FOs content taken as the investigation index.
The paper presents many interesting points, and the investigation into the selection of the resin and the best combination to increase the concentration of FOs is comprehensive and detailed. However, there are many inaccuracies in the text that hinder its correct interpretation, and some parts need to be detailed further for better flow and understanding. Therefore, in my opinion, the paper requires major revision.
Here are the specific points:
• Microorganism names should always be written in italics font; in the text, many names are not correctly formatted, not only for Hypsizigus marmoreus. Please correct it.
· Line 78: PDA should be potato dextrose agar? Please specify.
• Paragraph 2.2.1: Some sentences are written in the past tense, while others are in the present tense (e.g., mix). Please correct; typically, the past tense is used consistently.
• Paragraph 2.2.1: for all reagents, nutrients, etc., the source must be indicated. It is completely missing in this paragraph.
• Line 108, 117, 123, etc.: the subtitles should be better highlighted within the paragraph, for example by spacing them from the text or indicating them differently.
• Line 163: which monosaccharides are quantified?
• Line 248: This passage discusses the changes in feruloyl oligosaccharides (FOs) content in the fermentation broth over a period of 10 days. This passage needs further discussion; for example, why do FOs decrease after the sixth day?
• Line 273: pay attention to typos and incorrect spacing in this paragraph.
• Line 299: the optimum scheme is A4B5C4D3E4F3. How did you arrive at this conclusion? Please explain it in the text.
• Lines 299-300: we could see that FOs content of A4B5C4D3E4F3 and A4B5C4D3E4F3 was higher than other schemes. What does this mean?
• Line 308: in Table 4, it is indicated that for three experimental schemes A4B5C4D3E4F3 there are three different FOs results? What does this mean? Are they three replicates? If so, you cannot write that "under this condition, FOs concentration of rice bran fermentation broth was 2.38 mmol/L," but you must indicate the mean of the three values and the standard deviation.
• Line 310: "The characteristic peak time of ferulic acid was 12.516min in Fig. 3a, and that of ferulic acid was 12.541min in Fig. 3b." I see other retention times indicated in the peaks of the figures you mentioned. Please explain what you mean.
Reviewer 3 Report
Comments and Suggestions for Authors
The Manuscript ID: fermentation-2911810 “Preparation, purification, characterization and antioxidant activity of rice bran fermentation broth with Hypsizigus marmoreus” has an innovative purpose. However, the present manuscript needs "major revision" before being accepted for "Fermentation".
2. Materials and Methods
Line 78 - Describe the PDA culture medium. Also add, brand, city, and country.
Line 85 - mineral, distilled or deionized water?
Line 85 - hydrochloric acid (add concentration, brand, city, and country).
Line 87 - What is this culture medium? Add the brand, city, and country.
Add the brand, city and country for all chemical compounds and culture media in the entire "Material and Methods" topic. Please correct this.
3. Results
In Figure 1 - Write in the caption of this figure the meaning of the letters in the statistical analysis.
In Table 1, 2 and 3 - Write in the legend of these tables the meaning of the letters of the statistical analysis.
Figure 2 shows a lot of information. Authors need to separate these figures. Write the meaning of the letters in the statistical analysis in the caption of this figure.
4.Discussion
The discussion of this study is not adequate. The authors need to better discuss the study, making comparisons with related and current studies.
The authors do not describe the importance of the study carried out. This is important. It is also necessary to describe the scientific application of this study and the future perspective.
Comments on the Quality of English LanguageMinor editing of English language required.
Round 2
Reviewer 2 Report
Comments and Suggestions for Authors
Despite the discussion, I still don't understand why you write that two identical patterns (A4BC4D3E4F3 and A4BC4D3E4F3) are the optimal schemes. Wouldn't it be sufficient to write that A4BC4D3E4F3 is the optimal scheme? The same applies to the table 4, where there are three different results for three identical patterns. Please explain this passage better and why you repeat identical patterns in the text
Reviewer 3 Report
Comments and Suggestions for Authors
Dear authors
All of my recommendations were accepted and amended in the manuscript. I am pleased with the quality of the manuscript in its current version. I wish you all success.
Author Response
Thank you very much for your comment!